# Long COVID Prevalence and the Impact of the Third SARS-CoV-2 Vaccine Dose: A Cross-Sectional Analysis from the Third Follow-Up of the Borriana Cohort, Valencia, Spain (2020–2022)

**DOI:** 10.3390/vaccines11101590

**Published:** 2023-10-12

**Authors:** Salvador Domènech-Montoliu, Joan Puig-Barberà, Gema Badenes-Marques, María Gil-Fortuño, Alejandro Orrico-Sánchez, María Rosario Pac-Sa, Oscar Perez-Olaso, Diego Sala-Trull, Manuel Sánchez-Urbano, Alberto Arnedo-Pena

**Affiliations:** 1Emergency Service, University Hospital de la Plana, 12540 Vila-real, Spainmanu.msu@gmail.com (M.S.-U.); 2Vaccine Research Unit (AIV), Fundación para el Fomento de la Investigación Sanitaria y Biomédica de la Comunitat Valenciana (FISABIO), 46020 Valencia, Spain; alejandro.orrico@fisabio.es; 3Microbiology Service, University Hospital de la Plana, 12540 Vila-real, Spain; gil_marfor@gva.es (M.G.-F.); perez_oscola@gva.es (O.P.-O.); 4Centro de Salud Pública de Castellón, 12003 Castelló de la Plana, Spain; charopac@gmail.com (M.R.P.-S.);; 5Department of Health Science, Universidad Pública de Navarra, 31006 Pamplona, Spain

**Keywords:** long COVID, SARS-CoV-2 vaccination, booster dose, cellular immunity, prevalence, risk factors, human

## Abstract

Background: In March 2020, a COVID-19 outbreak linked to mass gathering dinners at the Falles Festival in Borriana, Spain, resulted in an estimated attack rate of 42.6% among attendees. Methods: In June 2022, we conducted a cross-sectional follow-up study of 473 adults aged 18 to 64 who attended the dinners at the Falles Festival in 2020, examining the cumulative experience after SARS-CoV-2 infection and vaccination responses. Data included demographic details, lifestyle habits, medical history, infection records, and vaccinations from a population-based vaccine registry. Blood samples were analyzed for SARS-CoV-2 antibodies and cellular immunity. We employed a doubly robust inverse-probability weighting analysis to estimate the booster vaccine dose’s impact on long COVID prevalence and symptom count. Results: A total of 28.1% of participants met the WHO criteria for long COVID, with older individuals showing higher rates. Long COVID diagnosis was less likely with factors including O blood group, higher occupational status, physical activity, three vaccine doses, strong SARS-CoV-2-S-reactive IFNγ-producing-CD8+ response, and infection during the Omicron period. Increased age, high or low social activity, underlying health conditions, a severe initial COVID episode, and reinfection were associated with higher long COVID likelihood. A booster dose, compared to one or two doses, reduced long COVID risk by 74% (95% CI: 56% to 92%) and symptom count by 55% (95% CI: 32% to 79%). Conclusion: Long COVID was prevalent in a significant portion of those who contracted COVID-19, underscoring the need for sustained follow-up and therapeutic strategies. Vaccinations, notably the booster dose, had a substantial beneficial effect on long-term infection outcomes, affirming the vaccination’s role in mitigating SARS-CoV-2 infection consequences.

## 1. Introduction

At the time of writing (July 2023) there have been 767,518,723 confirmed cases of COVID-19 globally, including 6,947,192 deaths, as reported to WHO [1]. Early in the pandemic, there were reports of persistent and prolonged effects after acute COVID-19 [2]. An unwelcome outcome of the SARS-CoV-2 pandemic has been non-recovery of health to that before the infection due to the persistence of symptoms in a substantial number of infected subjects [3,4,5,6]. This condition is known as long COVID [7,8].

Overall, the percent of COVID-19 cases with post-COVID symptoms has been estimated at a prevalence between 23 and 43% [6,9,10], and at 10–16% in vaccinated cases [4,10]. Most long COVID cases are in non-hospitalized patients with a mild acute illness [4,6].

The diagnosis and definition of post-acute sequelae of coronavirus disease 2019 or long COVID are an ongoing medical challenge that is not yet solved [11,12]. According to the World Health Organization (WHO) clinical case definition [13,14], the post-COVID-19 condition occurs in individuals with a history of probable or confirmed SARS-CoV-2 infection, usually three months from the onset of COVID-19, with symptoms that last for at least two months and cannot be explained by an alternative diagnosis. Long COVID is now recognized as a multi-organ disease with a broad spectrum of manifestations [4,11,15]. Common long COVID symptoms include anosmia and dysgeusia, fatigue, shortness of breath, cognitive dysfunction, hair loss, chest pain, cough, myalgia, and respiratory disorders that generally have an impact on everyday functioning [3,16].

Research looking at the impact of vaccination on existing long COVID has yielded varying results [10,17,18]. The benefit of COVID-19 vaccines on post-COVID-19 condition needs further study and gathering of evidence.

In March 2020 we had the opportunity to study an outbreak of 570 COVID-19 cases among 1338 persons attending mass gathering events in Borriana, a city in the Castellón province, Valencia Region, in Spain [19]. We conducted two follow-up studies of this cohort, the first in June 2020 [19] and the second in October 2020 [20]. In the second study we found that, six months after the infection, 33% of the COVID-19 cases complained of persistent symptoms [20]. The vaccination campaign started in Spain at the end of December 2020; by the end of June 2022, 97 million doses had been distributed in Spain, with a rate of vaccination in the population (one or two doses of vaccine as appropriate) of 86%, and 56% had received a booster dose [21].

In June 2022 we conducted a third follow-up study. At this time, we could observe the impact on the cohort of the exposure to the Alpha, Delta, and Omicron BA.1 and BA.2 variants, and to the SARS-CoV-2 vaccination that started during the first trimester of 2021. Our aim is to describe the prevalence of long COVID, associated factors, clinical manifestations, antibody and cellular immune responses, and vaccination impact of a third booster dose compared to two doses on the risk of long COVID in a selected subset of the Borriana cohort, two years after the pandemic onset.

## 2. Materials and Methods

### 2.1. Study Design

The origin of the Borriana COVID-19 cohort and the study methods have been described in previous publications [19,20,22,23]. In short, in March 2020 a COVID-19 outbreak took place during a series of mass gathering events involving 3200 participants at the Falles Festival in Borriana, a city of 34,000 inhabitants, in the Castellon province, Valencia Community (Spain). The outbreak resulted in 536 laboratory-confirmed cases of COVID-19 among 1132 subjects coming from an original stratified sample of 1663 participants [19]. We performed a second serological and epidemiological follow-up in October 2020 [20,22,23].

We describe here the third follow-up of the Borriana COVID-19 retrospective cohort study that we conducted in June 2022. We invited the 1132 subjects with valid laboratory test results in the first follow-up to participate [19]. Our primary objective was to study the prevalence and factors associated with long COVID in adults who attended the mass gathering events at the Falles Festival. The study population was restricted to cohort members who accepted our invitation, provided informed consent, were aged ≥ 18 and <65 years old, and had experienced at least one COVID-19 episode by June 2022 (Figure 1). While there is considerable debate about comparing post-COVID symptoms between those infected and those uninfected with SARS-CoV-2, we deemed it essential to explore differences specifically among the infected to provide a more focused understanding of the post-infection experience.

### 2.2. Ethics

The Research Ethics Committee of the Health Department La Plana University Hospital approved the study protocol (registry number 2961). All procedures were conducted according to good clinical epidemiological practices [24,25]. All participants provided signed informed consent to participate in the study.

### 2.3. Data Collection

Health staff from the University Hospital La Plana of Vila-real, the Public Health Centre of Castelló, and the health centers of Borriana, Vila-real, Onda, and La Vall d’Uixó obtained the informed consent, collected blood samples, completed the data collection form via face-to-face or phone interviews, and acquired information on vaccines administered (vaccine brand and vaccination dates), consulting the population-based Valencia Region Vaccine Information System. From the interview we obtained demographic data (age, sex), number of household members, occupational class and education levels, lifestyle (smoking habit, alcohol consumption, physical exercise, intensity of social contacts, a composite variable containing information on the frequency of travel outside one’s hometown, attendance at Falles association among locals, and the frequency of visiting restaurants and bars or work), body mass index (BMI = kg/m^2^, obesity ≥ 30.0), underlying chronic illnesses, and SARS-CoV-2 infection and reinfection information. For each infection episode, we obtained the date of infection, symptoms, duration of the illness, medical consultation, admission, and diagnostic tests performed. We also collected information on return to the usual state of health, health recovery, post-COVID sequelae, duration of the sequelae, and the presence of 31 symptoms at the time of the interview. Participants were asked to complete the Generalized Anxiety Disorder Screener (GAD-7) and Patient Health Questionnaire 9 (PHQ-9) in their validated Spanish versions [26,27,28,29]. At the time of the interview, researchers were unaware of long COVID status.

### 2.4. Laboratory Procedures

Laboratory procedures in the previous follow-up studies are described in a previous publication [30]. For the current follow-up, sera were tested via chemoluminiscence immunoassay (CLIA)) (Alinity i Anti-SARS-CoV-2 S and N, Abbot Laboratories, Chicago, IL, USA); the S assay targets the SARS-CoV-2 spike protein and the N assay targets antibodies to the nucleocapsid protein. We interpret anti-N as a proxy for previous SARS-CoV-2 infection [31]. We also determined the vitamin D sera levels and, if not already available, the blood group. All tests were performed at the Microbiology Service Laboratory of the University Hospital de la Plana, Vila-real.

In December 2022, we obtained additional blood samples from a subset of 225 cohort members to determine cell immunity against the Ancestral and Omicron BA.2 variants of concern (VOCs). Determinations were performed with flow cytometry functional cellular assays based on the detection of markers of T cell activation such as the IFN-ɣ release assay, QIAGEN’s QuantiFERON SARS-CoV-2 [32]. Enumeration of SARS-CoV-2-S-reactive IFNγ-producing-CD8^+^ and CD4^+^ T cells in fresh heparinized whole blood was carried out via flow cytometry for intracellular cytokine staining (BD Fastimmune, Becton Dickinson and Company-Biosciences, San Jose, CA, USA) as previously described [33,34] (Appendix A). Specimens were assayed at the Microbiology Service of the Hospital Clínico Universitario of Valencia.

### 2.5. Statistical Analysis

#### 2.5.1. Outcomes and Exposure Definitions

Definitions: SARS-CoV-2 infection, long COVID, and number of reported symptoms.

We defined SARS-CoV-2 infection by either recall or a previously positive laboratory test, regardless of symptoms. We defined long COVID in individuals with symptoms for at least two months and three months from the date of a previous SARS-CoV-2 infection [13,14]. We computed the number of symptoms per subject from the positive answers to the presence of any of 31 symptoms in the face-to-face interview.

#### 2.5.2. Infection Period

We considered four major time periods of SARS-CoV-2 infection as per the predominant VOC in Spain in each period [35,36]: Ancestral-D164G (any infection < 1 January 2021), Alpha (any infection ≥ 1 January 2021 and < 1 August 2021), Delta (infection ≥ 1 August 2021 and before 1 December 2021), and Omicron (infection ≥ 1 December 2021 to June 2022) (Figure 2).

#### 2.5.3. Priming and Booster

We defined subjects as primed 14 days after receiving one and two doses, and boosted 14 days after the third dose. Subjects that did not comply with the former criteria, not vaccinated, or with missing vaccination dates were excluded from the analysis of the booster effect on long COVID.

#### 2.5.4. Descriptive Analysis

We assessed differences by long COVID status in the distribution of sociodemographic factors, intensity of social contacts, leisure, or work, clinical manifestations, and immune status. We considered standardized mean differences (SMDs) ≥ 0.2 to indicate a non-negligible difference in the mean or prevalence of a covariate by long COVID status [37]. SMDs are not influenced by sample size and allow for the comparison of the relative balance of variables measured in different units [38]. We used the *t*-test, chi-squared (χ^2^) test, and non-parametric test for bivariate comparisons, and the trend test for ordinal variables; we considered a difference as significant if the bilateral *p*-value was <0.05. We assessed departure from linearity in categorical ordered variables, and interaction between potential confounders using the likelihood ratio test. We estimated the odds ratio (OR) associated with sociodemographic factors and other risk factors using unadjusted bivariate logistic regression.

#### 2.5.5. Inferential Analysis

To estimate the effect of the booster dose on the risk of long COVID and the number of symptoms per subject, we restricted our analysis to the subjects with a first infection during the period of the third dose rollout, January 2022 onward, and compared the risk of long COVID and the number of symptoms in boosted versus primed with one or two doses. To assess potential confounding due to healthy vaccinee bias and test the robustness of our findings, we conducted a supplementary sensitivity analysis. This analysis focused on individuals who experienced their first infection in March 2020, a period when the vaccine was not available (Figure 2). This approach aimed to evaluate the association between the booster dose and both the risk of long COVID and the number of symptoms, thereby testing our findings against possible unmeasured, unknown confounding factors. We estimated the booster impact as the adjusted treatment effect obtained from doubly robust inverse-probability weighting [39]. We used previous knowledge and directed acyclic graphs (DAGs) [13,14] to define the variables to improve comparability and exchangeability between the comparison groups and to clarify the minimum set of variables to perform the adjusted analysis (Appendix A). Following both criteria, we adjusted age to be a categorical variable, 18–24, 25–34, 35–49, 50–64; blood group to be a four-category variable; occupational class as qualified and over compared to other; household membership size in two categories, one or two, and two or more members; body mass index as ≥30 or lower; and outpatient consultation for the 1st episode, yes or no. The booster effect, as relative risk reduction (RRR), was estimated as the ((risk in those not boosted − risk in boosted group)/risk in those no boosted)) × 100. The RRR confidence intervals were computed using the delta method for possibly nonlinear combinations of parameter estimates [40]. We considered OR 95% confidence intervals to reject the null hypothesis of no effect when the unity value was not contained, and in prevalence or RRR when the 95% confidence interval did not include the 0 value.

#### 2.5.6. Statistical Package

We performed all analyses with STATA 18^®^ (StataCorp, 4905 Lakeway Drive College Station, TX, USA).

## 3. Results

### 3.1. Major Characteristics of the Included Subjects

We included 473 individuals aged 18–64 previously infected with SARS-CoV-2. The mean age of all subjects was 41.9 ± 12.9 (mean ± standard deviation). Females outnumbered males (300 vs. 173; 63.4% vs. 33.8%) (Table 1). Regarding other characteristics, 288 (62.9%) of the included individuals were at least qualified or skilled; 216 (46.6%) had a university education level; 262 (55.4%) reported regular physical activity; 190 (40.5%) were smokers or ex-smokers; 143 (30.6%) had underlying illness; and 133 (28.3%) were obese. Only 9 (1.9%) of all subjects had not received at least one vaccine dose; 315 (66.9%) had their first infection in the Ancestral period and 134 (28.5%) in the Omicron period (Figure 2 and Table 1); finally, 120 (25.4%) recollected a second COVID episode. The prevalence of long COVID was higher in those who were infected during the Ancestral period (99 out of 315; 31.4%) as compared to the Omicron period (23 out of 134; 17.2%), (*p* = 0.002).

### 3.2. Factors Associated with Long COVID

We found that 133 (28.1%) of the 473 included cohort members met the criteria for long COVID (Figure 1). We observed the prevalence of long COVID to increase with advancing age, with a proportion of 10.0% (7 out of 70) in the 18 to 24 age group, 29.4% (74 out of 252) in the 25 to 49 age group, and 34.4% (52 out of 151) in the 50 to 64 age group, with a significant trend for age (*p* = 0.0010). Long COVID patients were older than non-long COVID patients (44.5 ± 11.7 vs. 40.8 ± 13.2; SMD −0.30, not shown in Table 1). When comparing individuals with and without long COVID, we found that the O blood group, higher occupational class, more than two household members, regular physical activity, three vaccine doses, and infection in the Omicron period were related to a lower prevalence of long COVID. In addition to age, a minimal or high intensity of social contacts, underlying illnesses, and reinfection were related to a higher probability of long COVID (Table 1).

### 3.3. COVID Episode Characteristic and Long COVID Status

Long COVID was more prevalent among individuals who suffered a more severe initial COVID-19 episode in comparison to their counterparts who did not experience long COVID, as presented in Table 2. Long COVID patients, after their initial SARS-CoV-2 infection, reported a higher frequency of symptomatic episodes (90.2% vs. 72.8%), of longer duration (12.9 vs. 7.0 days), with more outpatient consultations (58.0% vs. 40.7%) and hospital admissions (6.8% vs. 1.8%), as compared to the cohort members who did not meet the criteria for long COVID definition. With regards to reinfection, it was observed that the presence of symptoms, medical consultations, and admissions were more common (SMD > 0.2) in long COVID patients; however, estimated OR CIs were nonsignificant (Table 2).

### 3.4. Post-COVID Outcomes

In total, 77.8% of the included members identified themselves as fully recovered following their SARS-CoV-2 infection. Furthermore, 81.4% perceived their current health status as either good or excellent, while 73.8% reported being as healthy as prior to their infection. A noteworthy proportion of individuals without long COVID demonstrated superior parameters across these three outcomes in comparison to subjects who experienced long COVID (Table 2). A significantly greater number of long COVID patients reported experiencing sequelae (84.2%) when contrasted with those without long COVID (9.7%). In addition, individuals with long COVID were at a significantly higher risk of anxiety (OR, 1.79; 1.18–2.72) or depression (OR, 1.82; 1.15–2.86) as compared to their counterparts without long COVID.

### 3.5. Reported Symptoms

At the time of the follow-up in June 2022, all participants were queried regarding the existence of a battery of 31 symptoms. The mean number of symptoms reported was 2.2 ± 3.22, with a notable significant difference in the number of symptoms reported between no long COVID and long COVID subjects, i.e., 1.6 ± 2.7 versus 3.8 ± 3.9, SMD −0.6550, *p* = 0.0001 (Table 2; *p*-value not shown). There was no observable variation in the number of symptoms reported in those with a singular COVID-19 infection compared to those with a reinfection (mean ± standard deviation): 2.1 ± 3.1 symptoms versus 2.7 ± 3.9, SMD −0.1747, *p* = 0.1043.

As illustrated in Figure 3 and Table 3, there was a noteworthy differentiation in the prevalence of 15 out of the 31 symptoms between individuals with long COVID and those without. The symptoms that exhibited the most substantial differences in absolute percentage points and were most frequently reported include fatigue/tiredness and hair loss (with a difference greater than 20% and present in 40.6% and 33.8% of long COVID subjects, respectively). Other symptoms with notable differences include dysgeusia, restlessness, dyspnea, memory loss, difficulty in concentration, anxiety, and insomnia. Although less frequent, visual flashes, speech difficulties, and nausea or vomiting were still considerably more prevalent in individuals with long COVID, with a standardized mean difference (SMD) greater than 0.2.

### 3.6. Vaccine Booster Dose Impact on Long COVID Prevalence and the Number of Reported Symptoms

After excluding nine non-vaccinated cases and three cases who were infected <14 days after vaccine administration, and with no further exclusions, 319 (69.2%) cohort members were immunized with a third dose or boosted, and 142 (30.8%) had received one or two vaccine doses.

When performing the analysis restricted to the 117 cohort members with their first infection during the Omicron period and on or after the 1 January 2022, the period of the third dose rollout (Figure 2), 90 (76.9%) were boosted and 27 (23.1%) were primed. The unadjusted prevalence of long COVID was 13.3% among boosted and 29.6% in those primed; SMD = 0.40. The number of symptoms was 1.5 ± 2.6 in boosted compared to 2.4 ± 3.7 in primed; SMD, 0.29.

In the adjusted analysis, we estimated an absolute reduction in the prevalence (%) of long COVID of −38.2% (−60.5% to −15.9%) among boosted subjects compared to primed subjects. Furthermore, an adjusted RRR of 74.0% (56.4% to 91.7%) was observed in boosted subjects compared to primed subjects. In terms of the number of symptoms, we noted that boosted subjects experienced a mean reduction of −1.8 (−3.3 to −0.4) symptoms when compared to primed subjects. The adjusted RRR in those who were boosted vs. primed was 55.2% (31.6% to 78.7%) (Table 4).

### 3.7. Sensitivity Analysis

To assess the sensitivity of our findings, we estimated the booster effect on the prevalence of long COVID and the number of symptoms experienced by individuals who had their first infection in March 2020, well before the vaccine became available. Of the 267 cohort members, a total of 188 (70.4%) were boosted, while 79 (29.6%) were primed. The unadjusted observed prevalence of long COVID was 40.5% in boosted subjects and 30.9% in primed subjects, which is indicative of a non-negligible difference, with an SMD of 0.20. In terms of the number of symptoms, the primed group reported 3.1 ± 3.9, while the boosted group reported 2.6 ± 3.4, with no evidence of a difference between the groups, as indicated by an SMD of 0.14. In the adjusted analysis, all the confidence intervals were compatible with either no significant booster effect or a null booster effect (Table 4).

## 4. Discussion

### 4.1. Summary of Key Findings

Our study, based on the Borriana COVID-19 cohort in the Valencia Region of Spain, provides valuable insights into the prevalence, risk factors, clinical manifestations, and vaccination impact in the number of symptoms and the likelihood of long COVID in adults aged 18 to 64 who had experienced at least one SARS-CoV-2 infection episode from March 2020 to June 2022.

Using the long COVID WHO definition [13,14], we observed that long COVID was present in one in three cohort members after their initial SARS-CoV-2 infection. Our findings suggest that increasing age, low social class and social support, underlying illnesses, reinfection, and severity of the first episode are associated with a higher risk of long COVID. Subjects with long COVID reported a poorer perception of their health status, experienced an average of four persistent symptoms compared to two in subjects without long COVID, and more frequently experienced moderate to severe anxiety or depression than those without long COVID. Notably, our data indicate that receiving a booster dose of the COVID-19 vaccine may be associated with a lower prevalence of long COVID and fewer reported symptoms.

### 4.2. Long COVID Definition and Comparison Groups

Our study suggest that long COVID is not a rare outcome, aligning with previously published estimates of ranges from 25% to 46% in non-hospitalized patients [6,9]. The higher prevalence of long COVID by age group in our cohort is consistent with other studies [41,42], also emphasizing the need for targeted interventions to prevent long COVID in the less than 65 years old age group. In addition to age, the association of long COVID with social factors, genetic profile, and underlying disorders observed in our study is also in line with the existing literature [4,6,41,43]. Although we did not identify a differential risk by gender, we cannot discard it, as we observed a 18% higher risk in females, albeit not reaching significance.

While there is evidence of prolonged immune activation in severe COVID-19 cases, the relationship between this immune activation and long COVID symptoms is complex and not directly correlated [44]. A number of studies considered parameters of antibody and T cell immunity to SARS-CoV-2 antigens, especially S and N, to evaluate the hypothesis that the risk of long COVID might be associated with an abnormally high or low immune response to acute infection [45]. We have observed that a SARS-CoV-2-S-reactive IFNγ-producing-CD8+ response was significantly associated with a lower risk of long COVID (Table 1). It has been shown that CD8+ cytotoxic lymphocytes interrupt viral replication by secreting antiviral cytokines (IFN-gamma and TNF-alpha) and directly killing infected cells, negatively correlating with stages of disease progression [46].

Previous infection with pre-Omicron variants is one of the risk factors for post-acute sequelae of SARS-CoV-2 symptoms [6]. The current estimates of the prevalence of post-COVID symptoms and functional impairment are mostly based on infection with early variants of SARS-CoV-2 [47]. As we also observed, some authors have described a lower risk of long COVID in subjects infected during the Omicron period, suggesting that infection by BA.1 and BA.2 lineages was of less severity than previous variants. It has been suggested that the better outcomes could be due to the immunity provided by previous infection and vaccination. We provide relevant information circumscribed to those infected in the Omicron period and show the salutary effect of booster doses of vaccines. Supporting this hypothesis, we found an inverse-probability-weighted adjusted prevalence of long COVID at the time of follow-up to be 51.6% (30.4% to 72.8%) in those who received only the primary vaccine series, compared to 13.4% (6% to 20.8%) in those who received a booster dose. We speculate that, although the overall burden of COVID has diminished, the possibility of adverse outcomes following SARS-CoV-2 infection with new circulating variants should not be discounted, especially among socially active adults who have not received updated vaccine protection or whose vaccine protection has waned.

### 4.3. Vaccination and Long COVID

Our findings contribute to the growing body of evidence on the potential benefits of COVID-19 vaccination in reducing the risk of long COVID [48,49], especially in those less than 65, as reported by others [50].

Recent systematic reviews conclude that vaccination before SARS-CoV-2 infection could reduce the risk of subsequent long COVID; however, the effect of vaccination on people with pre-existing long COVID was found to be inconsistent, with some studies showing symptom improvement and others not [18]. Vaccination with two doses could be more effective than a single dose in reducing the risk of developing long COVID [48]. Vaccination before Omicron BA.1 infection including the receipt of booster vaccines was not associated with a significant protective effect against the number of post-acute COVID-19 symptoms [51], whereas others report that symptoms were less common in vaccinated versus non-vaccinated individuals with Omicron infection [47]. Current studies do not report specific evidence on the impact of a booster or third dose of the COVID-19 vaccine on long COVID in patients who have already been infected [45,52].

Our results suggests that individuals who received a booster dose of the vaccine, and were infected during the Omicron period, reported fewer long COVID symptoms and a lower prevalence of long COVID compared to those who were primed. This is a significant finding, given the global push for booster vaccinations. Additionally, our sensitivity analysis, focusing on individuals with their first infection before the vaccine rollout, did not show a significant booster effect, supporting the validity of our findings regarding the positive vaccination impact.

### 4.4. Strengths of the Study

The major strengths of our study relate to the follow-up of the cumulative experience during more than two years of a well-defined and studied incipient cohort, with broad data collection, the use of a population-based vaccine registry, determination of markers of the immune response, and a robust statistical analysis. We argue, additionally, the relevance of comparing the outcomes among a socially active community of adults aged 18–64, previously infected with SARS-CoV-2 at the time of follow-up. The collection of data was unaffected by prior knowledge of the long COVID status of the cohort members included in the study; more than 85% of COVID episodes were laboratory confirmed, with no differences between subjects without or with long COVID criteria. The distribution of risk factors, episode characteristics, outcomes, and immune responses between the two groups was consistent with previously published systematic reviews [3,5,6,41,53]. Accordingly, we contend that our findings are pertinent for evaluating and interpreting the prevalence, risk factors, clinical outcomes, and vaccination impact on long COVID among the subjects in our cohort who were included in the June 2022 follow-up.

### 4.5. Limitations

Our study has several limitations. The cohort is based in a specific region of Spain, which limits the generalizability of our findings. Additionally, our study is observational, and while we employed a doubly robust inverse-probability weighting for our impact estimations, residual confounding may still be present. The reliance on self-reported symptoms and medical history may also introduce recall bias. We did not assess the progression of symptoms and long COVID at multiple, defined follow-up intervals, so we cannot estimate the total number of individuals who ever had long COVID. We opted for an approach that interprets data as the cumulative risks and protective effects of booster vaccination. We believe this information remains pertinent for public health policy and patients, even if it does not allow more detailed temporal analyses of the relationships between infection, vaccination, and long COVID. We cannot discard selection bias due to self-reference bias as suggested by the overrepresented female group, at 63.4%, although females were overrepresented in the first study (60.1%) and the second follow-up (62.2%). The overrepresentation of females has been also reported, however, in other observational studies [51], and in some cases was as high as 80%, whereas in others it was males who were overrepresented [54]. Finally, we did not conduct an analysis based on vaccine type, time elapsed since vaccination, or the timing of vaccination relative to infection. This was due to the short post-vaccination follow-up period and the limited sample size. Nonetheless, we can speculate that calendar time would act as a non-differential bias, potentially biasing our results toward the null hypothesis. This makes our findings on the cumulative impact of booster doses at the time of the follow-up particularly relevant. We did not include potential adverse vaccination effects. Adverse vaccination effects have the potential to skew study results, particularly if they result in differential uptake of booster doses. These effects may also be correlated with other variables of interest, such as age or pre-existing conditions, thereby confounding the study outcomes. Our sensitivity analysis partially accounts for this potential bias. Future studies may need to address these limitations to further refine our understanding. The current findings should be interpreted considering these limitations.

### 4.6. Clinical Implications

Our findings underscore the importance of comprehensive care for individuals recovering from COVID-19, particularly those with identified risk factors for long COVID. The potential benefit of booster vaccinations in reducing long COVID prevalence and the number of persistent symptoms, as suggested by our data, may inform and reinforce public health policies regarding vaccination strategies in the coming seasons.

## 5. Conclusions

In conclusion, our study provides important insights into the prevalence, risk factors, and clinical manifestations of long COVID in the Borriana cohort in Spain. Our findings suggest a potential protective effect of booster vaccinations against long COVID, even in those under 65 years of age. Our findings contribute to a better understanding of who is at higher risk of long COVID and provide additional evidence for the benefits of vaccination, allowing a more focused follow-up and management of COVID patients and targeted social support and public health interventions. As SARS-CoV-2 continues to circulate and evolve, ongoing research into long COVID, its risk factors, and its potential mitigation and therapeutic strategies, including vaccination, remains a global priority.

## Figures and Tables

**Figure 1 vaccines-11-01590-f001:**
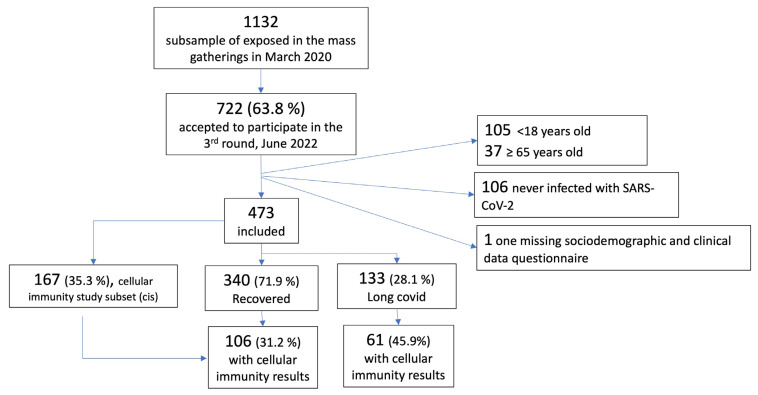
Study flowchart.

**Figure 2 vaccines-11-01590-f002:**
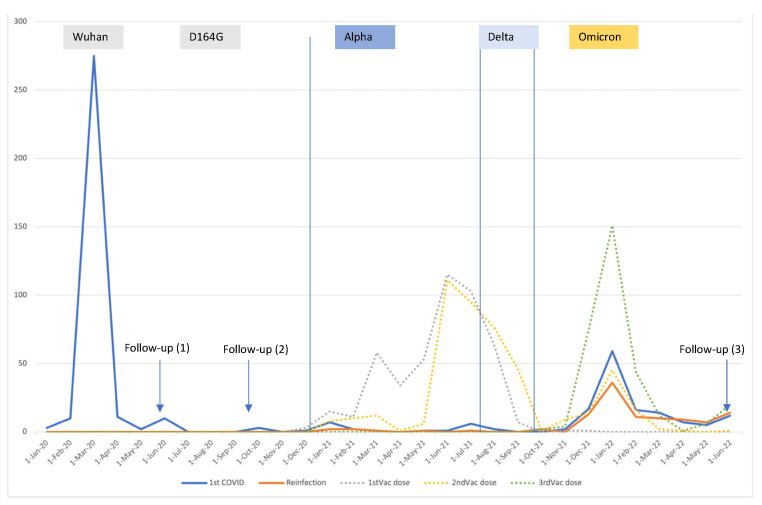
Timeline of first infection, reinfection, and vaccination (1st, 2nd, and 3rd doses) and VOC circulating periods for subjects included in the third follow-up study, June 2022. Borriana Cohort, 2020–2022.

**Figure 3 vaccines-11-01590-f003:**
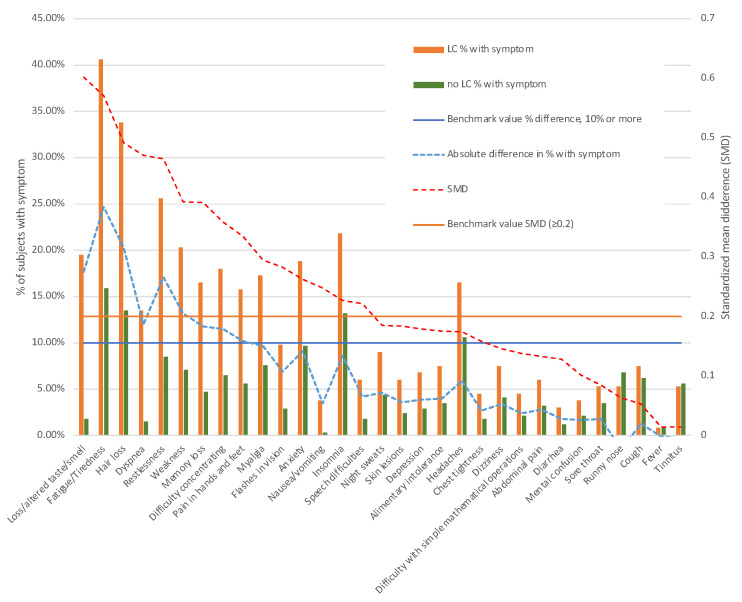
Symptoms post-COVID and their distribution by long COVID status.

**Table 1 vaccines-11-01590-t001:** Factors associated with long COVID.

	Long COVID						
	No	Yes	Total	SMD *	OR *	95%CI *	
	340 (71.9%)	133 (28.1%)	473 (100.0%)				
Age in groups in years							
18–24	63 (18.5%)	7 (5.3%)	70 (14.8%)	**0.43**	1.00		
25–49	178 (52.4%)	74 (55.6%)	252 (53.3%)		**3.74**	**1.64**	**8.55**
50–64	99 (29.1%)	52 (39.1%)	151 (31.9%)		**4.73**	**2.02**	**11.06**
Female	212 (62.4%)	88 (66.2%)	300 (63.4%)	0.08	1.18	0.78	1.80
Blood group							
O	156 (46.0%)	46 (34.8%)	202 (42.9%)	**0.28**	**0.39**	**0.20**	**0.79**
A	147 (43.4%)	63 (47.7%)	210 (44.6%)		0.57	0.29	1.13
B	24 (7.1%)	18 (13.6%)	42 (8.9%)		1.00		
AB	12 (3.5%)	5 (3.8%)	17 (3.6%)		0.56	0.17	1.86
Occupation: Qualified or over	218 (66.3%)	70 (54.3%)	288 (62.9%)	**0.25**	**0.60**	**0.40**	**0.91**
Education level University	164 (49.1%)	52 (40.0%)	216 (46.6%)	0.18	0.69	0.46	1.04
Household members > 2	276 (82.1%)	93 (70.5%)	369 (78.8%)	**0.28**	**0.52**	**0.33**	**0.83**
Social contacts							
Minimal	85 (25.1%)	39 (29.3%)	124 (26.3%)	**0.36**	**2.00**	**1.16**	**3.47**
Active	131 (38.6%)	30 (22.6%)	161 (34.1%)		1.00		
High	123 (36.3%)	64 (48.1%)	187 (39.6%)		**2.27**	**1.38**	**3.74**
Activity in contact with people	273 (80.8%)	100 (75.8%)	373 (79.4%)	0.12	0.74	0.46	1.20
Regular physical activity	198 (58.2%)	64 (48.1%)	262 (55.4%)	**0.20**	**0.67**	**0.44**	**1.00**
Somoker or ex-smoker	129 (38.3%)	61 (46.2%)	190 (40.5%)	0.16	1.39	0.92	2.08
Alcohol consumption (light or moderate)	260 (76.5%)	93 (69.9%)	353 (74.6%)	0.15	0.72	0.46	1.12
Underlying illnesses	94 (27.8%)	49 (37.7%)	143 (30.6%)	**0.21**	**1.57**	**1.02**	**2.41**
Obesity BMI ≥ 30	91 (27.0%)	42 (31.6%)	133 (28.3%)	0.10	1.25	0.81	1.93
Vit D blood levels 30 UI and over	139 (40.9%)	52 (39.1%)	191 (40.4%)	0.04	0.93	0.62	1.40
Number of COVID19 vacc doses							
No vaccinated	6 (1.8%)	3 (2.3%)	9 (1.9%)	**0.32**	0.80	0.19	3.36
One dose	9 (2.6%)	3 (2.3%)	12 (2.5%)		0.54	0.14	2.07
Two doses	82 (24.1%)	51 (38.3%)	133 (28.1%)		1.00		
Three doses	243 (71.5%)	76 (57.1%)	319 (67.4%)		**0.50**	**0.33**	**0.78**
Third dose vs one or two previous doses							
Primed	90 (27.0%)	52 (40.6%)	142 (30.8%)	**0.29**	1.00		
Boosted 3rd dose	243 (73.0%)	76 (59.4%)	319 (69.2%)		**0.53**	**0.35**	**0.83**
COVID cases by Predominant VOC circulation in Spain							
Ancestral	216 (63.9%)	99 (74.4%)	315 (66.9%)	**0.40**	1.00		
Alpha	9 (2.7%)	9 (6.8%)	18 (3.8%)		2.18	0.84	5.66
Delta	2 (0.6%)	2 (1.5%)	4 (0.8%)		2.18	0.30	15.71
Omicron	111 (32.8%)	23 (17.3%)	134 (28.5%)		**0.45**	**0.27**	**0.75**
Reinfected	77 (22.6%)	43 (32.3%)	120 (25.4%)	**0.22**	**1.63**	**1.05**	**2.54**
Cell immune response (IFN-γ-producing CD4+ or CD8+ T) † Median (p25–p75)							
CD4 Ancestral	0.9 (0.0–2.2)	0.7 (0.4–2.5)	0.8 (0.2–2.3)	0.12	ne	ne	ne
CD8 Ancestral	1.0 (0.0–3.4)	1.0 (0.4–2.3)	1.0 (0.0–2.8)	**0.44**	ne	ne	ne
CD4 BA.2	0.7 (0.0–2.5)	1.0 (0.4–3.9)	0.9 (0.1–2.8)	−0.19	ne	ne	ne
CD8 BA.2	0.6 (0.0–2.2)	1.0 (0.3–2.5)	0.8 (0.0–2.5)	0.09	ne	ne	ne
Anti SARS-CoV-2 antibody levels					ne	ne	ne
Anti-S AU: mean ± sd	2094.7 ± 1614.8	1853.6 ± 1485.0	2026.9 ± 1581.5	0.16	ne	ne	ne
Anti-N AU: mean ± sd	2.4 ± 2.8	2.4 ± 2.8	2.4 ± 2.8	0.03	ne	ne	ne

SMD (standardized mean differences) ≥ 0.2 to indicate a nonnegligible difference in the mean or prevalence of a covariate by long COVID status. OR, odds ratio. CI, confidence interval. * Non negligible difference or CI not containing the unity highlighted in bold. † Determined in a subsample of 167 (Figure 1). Number of SARS-CoV-2-reactive IFN-γ-producing CD4+ or CD8+ T cells relative to the absolute number of CD4+ and CD8+ T cells, respectively, ×100 (%). Any frequency value of SARS-CoV-2-reactive IFN-γ-producing CD4+ or CD8+ T cells after background substraction was considered as a positive (detectable) result and used for analysis purposes. The % is represented ×1000 to adapt to only one decimal position. S, spike protein. N, nucleocapsid. AU, arbitrary units. sd, standard deviation; ne, not estimated.

**Table 2 vaccines-11-01590-t002:** Characteristics of SARS-CoV-2 infection and its associated outcomes, both in aggregate and stratified by long COVID status.

	Long COVID			SMD *	OR *	95%CI *	
	No	Yes	Total				
	340 (71.9%)	133 (28.1%)	473 (100.0%)				
**Episode characteristics**							
**First episode**							
Symptomatic	246 (72.8%)	120 (90.2%)	366 (77.7%)	**0.46**	**3.45**	**1.86**	**6.42**
Laboratory confirmed	288 (84.7%)	120 (90.2%)	408 (86.3%)	0.17	1.67	0.88	3.17
Duration in days; mean ± sd	7.0 ± 11.6	12.9 ± 21.7	8.7 ± 15.4	**−0.34**	**1.02**	**1.01**	**1.04**
Outpatient consultation	138 (40.7%)	76 (58.0%)	214 (45.5%)	**0.35**	**2.01**	**1.34**	**3.03**
Admission	6 (1.8%)	9 (6.8%)	15 (3.2%)	**0.25**	**4.05**	**1.41**	**11.61**
**Second episode**							
Symptomatic	59 (77.6%)	37 (86.0%)	96 (80.7%)	**0.22**	1.78	0.64	4.91
Laboratory confirmed	69 (89.6%)	38 (88.4%)	107 (89.2%)	0.04	0.88	0.27	2.88
Duration in days; mean ± sd	4.11 ± 4.40	4.67 ± 3.46	4.30 ± 4.09	−0.14	1.03	0.94	1.13
Outpatient consultation	33 (43.4%)	23 (53.5%)	56 (47.1%)	**0.20**	1.50	0.71	3.18
Admission	0 (0.0%)	1 (2.3%)	1 (0.8%)	**0.22**	1.77	0.05	∞
**Outcomes**							
Fully recovered after covid	327 (96.2%)	41 (30.8%)	368 (77.8%)	**1.85**	0.02	0.01	0.03
Perceived Current Health Status							
Poor	9 (2.7%)	2 (1.5%)	11 (2.3%)	**0.54**			
Fair	37 (10.9%)	37 (28.2%)	74 (15.7%)				
Good	218 (64.3%)	80 (61.1%)	298 (63.4%)				
Excellent	75 (22.1%)	12 (9.2%)	87 (18.5%)				
As healthy as before	297 (87.4%)	52 (39.1%)	349 (73.8%)	**1.16**	0.09	0.06	0.15
Sequelae	33 (9.7%)	112 (84.2%)	145 (30.7%)	**2.24**	49.62	27.55	89.36
Days with symptoms; mean ± sd	113.80 ± 237.29)	594.61 ± 287.79	560.98 ± 309.35	**−1.82**			
Number of symptoms reported at follow-up	1.6 ± 2.7	3.8 ± 3.9	2.2 ± 3.22	**−0.66**	**1.22**	**1.14**	**1.30**
Scoring GAD-7 Anxiety Severity							
0–4; minimal anxiety	246 (72.4%)	79 (59.4%)	325 (68.7%)	**0.32**	1.00		
5–9; mild anxiety	59 (17.4%)	29 (21.8%)	88 (18.6%)		1.53	0.92	2.55
10–14; moderate anxiety	25 (7.4%)	14 (10.5%)	39 (8.2%)		1.74	0.86	3.52
15–21; severe anxiety	10 (2.9%)	11 (8.3%)	21 (4.4%)		**3.43**	**1.40**	**8.37**
GAD-7 score: Anxiety, mild to severe	94 (27.6%)	54 (40.6%)	148 (31.3%)	**0.28**	**1.79**	**1.18**	**2.72**
Depression severity measure							
No or minimal	273 (80.3%)	92 (69.2%)	365 (77.2%)	**0.31**	1.00		
Mild	47 (13.8%)	24 (18.0%)	71 (15.0%)		1.52	0.88	2.61
Moderate	14 (4.1%)	8 (6.0%)	22 (4.7%)		1.70	0.69	4.17
Moderate to Severe	4 (1.2%)	7 (5.3%)	11 (2.3%)		**5.19**	**1.49**	**18.14**
Severe	2 (0.6%)	2 (1.5%)	4 (0.8%)		2.97	0.41	21.37
PHQ-9 score: Depresion Mild to severe	67 (19.7%)	41 (30.8%)	108 (22.8%)	**0.26**	**1.82**	**1.15**	**2.86**

SMD (standardized mean differences) ≥ 0.2 to indicate a nonnegligible difference in the mean or prevalence of a covariate by long COVID status. OR, odds ratio. CI, confidence interval. GAD-7, Generalized Anxiety Disorder Screener. PHQ-9, Patient Health Questionnaire. * Non negligible difference or CI not containing the unity highlighted in bold.

**Table 3 vaccines-11-01590-t003:** Symptoms reported by subjects who met the criteria for long COVID and those who did not.

Symptom*	Long COVID		SMD	Absolute %Difference
	Yes (n = 133)	No (n = 340)		
Fatigue/Tiredness	40.6%	15.9%	0.57	24.7%
Hair loss	33.8%	13.5%	0.49	20.3%
Loss/altered taste/smell	19.5%	1.8%	0.60	17.7%
Restlessness	25.6%	8.5%	0.47	17.1%
Weakness	20.3%	7.1%	0.39	13.2%
Dyspnea	13.5%	1.5%	0.47	12.0%
Memory loss	16.5%	4.7%	0.39	11.8%
Difficulty concentrating	18.0%	6.5%	0.36	11.5%
Pain in hands and feet	15.8%	5.6%	0.33	10.2%
Myalgia	17.3%	7.6%	0.30	9.7%
Anxiety	18.8%	9.7%	0.26	9.1%
Insomnia	21.8%	13.2%	0.23	8.6%
Flashes in vision	9.8%	2.9%	0.28	6.9%
Headaches	16.5%	10.6%	0.17	5.9%
Night sweats	9.0%	4.4%	0.18	4.6%
Speech difficulties	6.0%	1.8%	0.22	4.2%
Alimentary intolerance	7.5%	3.5%	0.18	4.0%
Depression	6.8%	2.9%	0.18	3.9%
Skin lesions	6.0%	2.4%	0.18	3.6%
Nausea/vomiting	3.8%	0.3%	0.25	3.5%
Dizziness	7.5%	4.1%	0.15	3.4%
Abdominal pain	6.0%	3.2%	0.13	2.8%
Chest tightness	4.5%	1.8%	0.16	2.7%
Difficulty with simple mathematical operations	4.5%	2.1%	0.14	2.4%
Diarrhea	3.0%	1.2%	0.13	1.8%
Sore throat	5.3%	3.5%	0.08	1.8%
Mental confusion	3.8%	2.1%	0.10	1.7%
Cough	7.5%	6.2%	0.05	1.3%
Fever	0.8%	0.9%	0.01	−0.1%
Tinnitus	5.3%	5.6%	0.01	−0.3%
Runny nose	5.3%	6.8%	0.06	−1.5%

**Table 4 vaccines-11-01590-t004:** Impact of the booster dose on long COVID prevalence and the number of symptoms during the Omicron (vaccine rollout accomplished) and Ancestral (pre-vaccine sensitivity analyisis) periods. Borriana Cohort Study, March 2020–June 2022.

Period/Outcome	Observed					Adjusted IPW *			
	Primed		Boosted		SMD	Absolute Difference Boosted vs. Primed		Relative Reduction in Boosted vs. Primed	
	n or mean	% or sd	n	%		% or mean	95%CI	%	95%CI
Infected in the Omicron period (n; row %)	27	23.1%	90	76.9%					
Long COVID prevalence (n; column %)	8	29.6%	12	13.3%	0.40	−38.2%	(−60.5% to −15.9%)	74.0%	(56.4% to 91.7%)
Symptoms (mean ± sd)	2.4	±3.70	1.5	±2.55	0.29	−1.8	(−3.3 to −0.4)	55.2%	(31.6% to 78.7%)
Infected in the Ancestral period (n; row %)	79	29.6%	188	70.4%					
Long COVID prevalence (n; column %)	32	40.5%	58	30.9%	0.20	−10.4%	(−23.0% to 2.2%)	24.9%	(−49.8% to 0.0%)
Symptoms (mean ± sd)	3.1	±3.9	2.6	±3.41	0.14	−0.7	(−1.7 to 0.3)	21.0%	(−46.2% to 4.2%)

* Adjusted by age, as a categorical variable, 18–24, 25–34, 35–49, 50–64; blood group, as four categories; occupational class, qualified or more and other; household membership size in two categories, one or two members, and two or more members; body mass index as ≥30 or lower; and outpatient consultation for the 1st episode, yes or no. IPW doubly robust inverse-probability-weighted adjustment. Primed: ≥14 days received one or two doses. Boosted: ≥14 days after a third dose. SMD (standardized mean differences) ≥ 0.2 to indicate a nonnegligible difference in the mean or prevalence of a covariate by outcome. CI confidence interval.

## Data Availability

Available from the authors upon reasoned request.

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
