# Peer review of "Long COVID Prevalence and the Impact of the Third SARS-CoV-2 Vaccine Dose: A Cross-Sectional Analysis from the Third Follow-Up of the Borriana Cohort, Valencia, Spain (2020–2022)"

_vaccines, 2023, doi:10.3390/vaccines11101590_

Round 1

Reviewer 1 Report

Comments are embedded in the attached document. Overall, this is important work. Given this, it is important that the methods be sound. Therefore, please consider revising the analysis to present robust findings.

Review by a writer-editor will resolve minor issues.

Author Response

Please find our  point-by-point response to the reviewer’s comments n the attached file.

Reviewer 2 Report

Well designed and documented study to assess long Covid due to SARS CoV2 vaccinated individuals with one or two boosters vs non vaccinated individuals. Some interesting associations with long covid have been described in their findings. This will be of importance in addressing long Covid among patients.

I have two minor queries:

1.How was Omicron variant established in those in the early period of Covid?

(L 343-345)

2. Any reason why long covid was less in households with more than two  members? (Table 1)

Author Response

Please find our point-by-point response to the reviewer’s comments in the attached file

Reviewer 3 Report

It is currently critical to understand the value of vaccination in prevention long COVID, as new vaccines are being rolled out this month, and stakeholders must decide policies (including risks and benefits) for administering/receiving these vaccines. Therefore, the topic is timely, of immense interest, and urgent. This is an important paper with impacts on public health and should be published as soon as possible. That need for speed should be balanced against the need to correct technical/writing errors, which I enumerate below.

MAJOR ISSUE

The phrase "socially engaged" appears only three times in the manuscript. It is never defined precisely. And it is never explained in detail up front why the study focuses on these individuals, despite the emphasis in the Abstract that this is what the study is about. What fraction of the whole Spanish population is "socially engaged"? If "socially engaged" is merely a code phrase for "attended the Falles Festival" then use  "attended the Falles Festival" rather than socially engaged. There could be a lot people who attended the Falles Festival who are not socially engaged.

Please include some comments, discussion, and/or analysis on the difference between "ever had long COVID" and "have long COVID at the time of the third f/u assessment". Your paper focuses exclusively on "have long COVID at the time of the third f/u assessment", as best I can understand it. Can you provide an estimate of how many people in this cohort had long COVID at some point in time, but that resolved prior to the third f/u? If not, please say so, and write something like you "cannot estimate the total number of individuals who ever had long COVID" as one of your study limitations.

MINOR ISSUES

Figure 1

Very oddly formatted, as if the original drawing did not render. Please send a high-quality PDF or the original Figure to enable review.

Figure 2

Some of the thin blue lines go off the top of the chart. As I understand it, these demarcate the predominant VOC epochs. You can remove these lines, as these are distracting, appear to be data, and in any case the exact dates of epochal change are fuzzy.

ABSTRACT

"In March 2020, a COVID-19 outbreak associated with the Falles Festival in Borriana, Spain, led to significant infections."

Do not use the word 'significant' in scientific writing except to mean that a p-value is < 0.05.

Rather write in the exact (or approximate number), so write, e.g.

"In March 2020, a COVID-19 outbreak associated with the Falles Festival in Borriana, Spain, led to 10,000-15,000 infections."

"we conducted a cross-sectional follow-up study of 473 socially engaged adults aged 18 to 64"

Why is it important to call out "socially engaged"? Is there a reason to exclude the others? Is there something critical about your hypothesis that requires you only focus on socially engaged?

"doubly robust inverse weighted probability analysis"

is jargon. Try Googling it; <10 hits, most of which are yours. Please use words the global community can understand.

"Not complying with long COVID criteria associated with factors..."

do you just mean

"Long COVID diagnosis was less likely with factors..."

Please re-word your sentence so it is understandable.

"Conversely, ..."

This word doesn't mean what you think it means. Just delete it; your sentence will be fine.

"COVID risk by 74% (95% CI: 56.4% to 91.7%) and symptom count by 55.2% (95% CI: 31.6% to 78.7%)."

too many significant figures

just write

"COVID risk by 74% (95% CI: 56% to 92%) and symptom count by 55% (95% CI: 32% to 79%)."

INTRODUCTION

"At the time of writing"

we don't know when that was, please write exact date

"Patients themselves call it “long COVID”"

not all do, and many MDs do as well, maybe just write

"This condition is also known as long COVID."

"prevalence between of 43% and 23%"

flip, so

"prevalence between of 23% and 43%"

"long-COVID"

don't hyphenate, so

"long COVID"

"on existing Long COVID"

not capitalized, so

"on existing long COVID"

check entire document - multiple such errors

"Alfa"

in English,

"Alpha"

"While there's"

formal writing, so

"While there is"

consider calling the Wuhan strain something else. Try to avoid place-names in viral designations (e.g., "Spanish Flu" is no longer a cool phrase).

TEXT

"long Covid"

->

long COVID

vs.

no need to abbreviate in main text, so

versus

"SARSCoV-2"

->

SARS CoV-2

Sensitivity analysis

"To assess the sensitivity of our findings, we estimated the booster effect on the prevalence of long COVID and the number of symptoms experienced by individuals who had their first infection in March 2020, well before the vaccine became available."

What are you trying to convey in this section? The first sentence in this paragraph (quoted above) has nothing to do with a "sensitivity analysis".

Are you trying to state the the metric "number of symptoms" is sensitive to whether or not someone has long COVID? Or that it can differentiate the vaccinated form the unvaccinated?

ok

Author Response

Please find our point-by-point response to the reviewer’s comments  in the attached file. 

Round 2

Reviewer 1 Report

Edits improved the paper substantially. Thank you for your work and this important contribution to science and our understanding of long COVID.

No comments. Minor edits that can be addressed by the journal writer-editor.